# Modulating supramolecular binding of carbon dioxide in a redox-active porous metal-organic framework

Zhenzhong Lu[1], Harry G.W. Godfrey[1], Ivan da Silva[2], Yongqiang Cheng[3], Mathew Savage[1], Floriana Tuna[1], Eric J.L. McInnes[1], Simon J. Teat[4], Kevin J. Gagnon[4], Mark D. Frogley[5], Pascal Manuel[2], Svemir Rudić[2], Anibal J. Ramirez-Cuesta[3], Timothy L. Easun[6], Sihai Yang[1] & Martin Schröder[1]

Hydrogen bonds dominate many chemical and biological processes, and chemical modification enables control and modulation of host–guest systems. Here we report a targeted modification of hydrogen bonding and its effect on guest binding in redox-active materials. MFM-300(V$^{III}$) {[V$^{III}_2$(OH)$_2$(L)], LH$_4$ = biphenyl-3,3′,5,5′-tetracarboxylic acid} can be oxidized to isostructural MFM-300(V$^{IV}$), [V$^{IV}_2$O$_2$(L)], in which deprotonation of the bridging hydroxyl groups occurs. MFM-300(V$^{III}$) shows the second highest CO$_2$ uptake capacity in metal-organic framework materials at 298 K and 1 bar (6.0 mmol g$^{-1}$) and involves hydrogen bonding between the OH group of the host and the O-donor of CO$_2$, which binds in an end-on manner, OH$\cdots$O$_{CO_2}$ = 1.863(1) Å. In contrast, CO$_2$-loaded MFM-300(V$^{IV}$) shows CO$_2$ bound side-on to the oxy group and sandwiched between two phenyl groups involving a unique O$_{CO_2}\cdots$c.g.$_{phenyl}$ interaction [3.069(2), 3.146(3) Å]. The macroscopic packing of CO$_2$ in the pores is directly influenced by these primary binding sites.

[1] School of Chemistry, University of Manchester, Oxford Road, Manchester M13 9PL, UK. [2] ISIS Facility, STFC Rutherford Appleton Laboratory, Chilton, Oxfordshire OX11 0QX, UK. [3] The Chemical and Engineering Materials Division (CEMD), Neutron Sciences Directorate, Oak Ridge National Laboratory, Oak Ridge, Tennessee 37831, USA. [4] Advanced Light Source, Lawrence Berkeley National Laboratory, Berkeley, California 94720, USA. [5] Diamond Light Source, Harwell Science Campus, Oxfordshire OX11 0DE, UK. [6] School of Chemistry, Cardiff University, Cardiff CF10 3XQ, UK. Correspondence and requests for materials should be addressed to S.Y. (email: Sihai.Yang@manchester.ac.uk) or to M.S. (email: M.Schroder@manchester.ac.uk).

There is increasing interest in using porous materials for substrate binding, separation, storage, delivery and catalysis owing to their specific host–guest binding interactions[1,2]. Gaining molecular insights into these binding interactions is very important, but experimentally challenging. Firstly, the nature of host–guest binding is often within the regime of low-energy supramolecular contacts, and thus the guest molecules have great mobility in the pore and can be highly disordered[3,4]. For the study of hydrogen bonds in particular, the identification of structural and dynamic details has only been achieved via neutron scattering in exceptional cases[5]. Secondly, the porous nature of host materials leads to additional impediments for the in-depth investigation of host–guest systems, mostly due to the intrinsic structural disorder and overall lack of long-range order for certain types of porous materials (e.g., carbons, porous silica, large-pore zeolites)[6,7]. Design of a suitable host platform is the key to overcome these barriers to gain molecular information on the host–guest binding based upon multiple supramolecular contacts, particularly for hydrogen bonds.

Metal-organic frameworks (MOFs) are an emerging class of porous materials showing highly crystalline structures and great promise for gas adsorption[8,9]. Built from metal ions/clusters and organic ligands, MOFs often exhibit high surface areas (up to $\sim 7,000\,m^2\,g^{-1}$) and, more importantly, tunable functional pore environments[10–12]. MOFs can readily incorporate desired functionality by judicious choice of metal nodes and/or organic linkers[13–15]. Post-synthetic modification, broadly defined as chemical derivatization of MOFs after their formation, can afford MOFs with additional functionalities that cannot be obtained by routine synthesis[16,17]. Fine-tuning of the supramolecular host–guest binding requires precise chemical modification to the functional groups on the MOF backbones. To govern the formation of hydrogen bonds in particular, such modification, involving removal/addition of protons, will trigger concurrent redox reactions to occur within the MOF hosts. Post-synthetic redox modification on MOFs without destroying/changing the pore structure, especially through a single-crystal to single-crystal transformation, rarely succeeds because of the changes in bond interactions which leads to collapse of the framework[18–20]. Redox modifications of MOFs have been reported for systems containing $Fe^{II/III}$, $V^{II/III/IV}$, $Ti^{II/III/IV}$, $Cu^{I/II}$ and $Cr^{II/III}$ centres[21,22]. However, none of these MOF systems affords a pair of structural analogues ($M^{n+}$ and $M^{n+1}$-based MOF) as a platform for the direct investigation of guest binding. For example, MOF-74(Fe) features a hexagonal array of one-dimensional channels lined with co-ordinatively unsaturated $Fe^{II}$ centres[23]. The oxygen adsorption isotherm indicates that $Fe^{II}$ centres bind to $O_2$ preferentially over $N_2$, which is attributed to a partial electron charge transfer from $Fe^{II}$ to $O_2$. However, the $Fe^{III}$-based MOF analogue cannot be isolated readily because, upon desorption of $O_2$, the charge transfer is reversible.

We report herein the synthesis of single crystals of a $V^{III}$-based MOF, MFM-300($V^{III}$) and its $V^{IV}$ isostructural counterpart, MFM-300($V^{IV}$), which was obtained by oxidation of the $V^{III}$ precursor via a single-crystal to single-crystal transformation. These two materials show the same overall structural features with some differences in metal–ligand bond distances, but the main difference is conversion of V–(OH)–V in MFM-300($V^{III}$) to V–O–V linkers in MFM-300($V^{IV}$). This provides a unique platform to investigate the precise role of the OH group in the supramolecular binding of guest molecules. The binding of $CO_2$ has been thoroughly investigated in these two MOF systems via neutron diffraction and inelastic scattering, coupled with modelling. These experiments find that, surprisingly, the hydroxyl protons play a dominating role in the global binding of adsorbed $CO_2$ molecules in the pore, rather than just providing specific binding interactions to the localized $CO_2$ molecules nearby, as typically found in amine-functionalized MOFs[24]. Thus, as a result of the presence/absence of the hydroxyl proton, the binding and packing pattern for adsorbed $CO_2$ molecules in these two iso-structural MOFs are distinct. This study represents a unique analysis of the structural and dynamic characterization of hydrogen bonds (and other associated supramolecular interactions) that are stabilizing host–guest systems in the solid state.

## Results

**Synthesis and crystal structure of MFM-300($V^{III}$).** An oxygen-free hydrothermal reaction of $VCl_3$, $H_4L$ ($H_4L$ = biphenyl-3,3′,5,5′-tetracarboxylic acid), HCl and water in the molar ratio of 6:1:1.5:1,600 at 210 °C for 3 days afforded MFM-300($V^{III}$)-solv, $[V_2(OH)_2(L)]\cdot 6H_2O$, as a green microcrystalline material with a yield of ca. 90%. Single crystals of MFM-300($V^{III}$)-solv were obtained under acidic conditions ($VCl_3$:$H_4L$:HCl:water = 16:1:53:11,000), which afforded dark green prism shaped crystals ($0.5 \times 0.2 \times 0.2$ mm; yield < 5%) along with a large amount of re-crystallized organic ligand. It is likely that this highly acidic hydrothermal environment slows down the crystallization process of the MOF leading to slow growth of crystal nuclei into large crystals. Interestingly, synthesis of vanadium-based MOFs, especially in the single crystal form, has been very rarely reported[25]. It is also worth noting that oxygen-free conditions are necessary to prevent oxidation of $V^{III}$ centres during the hydrothermal process, and MFM-300($V^{III}$)-solv cannot be obtained in the presence of air/oxygen.

MFM-300($V^{III}$) crystallizes in the space group $I4_122$ and is isostructural to MFM-300(M) (M = Al, Ga, In, see Supplementary Table 1 for single crystal data)[26–28]. The $V^{III}$ centre shows octahedral coordination defined by four carboxylate oxygen atoms from the ligand [V–O = 2.014(2) and 2.030(2) Å] and two oxygen atoms from two hydroxyl groups [V–O = 1.948(2) Å] (Fig. 1). Adjacent pairs of $V^{III}$ centres are bridged by two carboxylates and a $\mu_2$-hydroxyl group forming an extended chain of $[V_2(OH)_2O_4]_\infty$ along the c axis. The ligands further bridge the vanadium hydroxyl chains to give a three-dimensional network with square-shaped channels decorated with hydroxyl groups. The cross-section of the channel is $6.7 \times 6.7$ Å taking Van der Waals radii into consideration. Guest water molecules in the channel can be readily removed by heating at 150 °C under vacuum for 5 h to give the desolvated material, MFM-300($V^{III}$), as studied by in situ single crystal diffraction, which confirmed the retention of the framework structure and negligible residual electron density within the pore. The phase purity of MFM-300(VIII)-solv was confirmed by PXRD (Supplementary Fig. 1).

**Post-synthetic oxidation to synthesize MFM-300($V^{IV}$).** The as-synthesized single crystals of MFM-300($V^{III}$) changed colour from green to dark purple by heating at 150 °C under a flow of $O_2$ for 16 h to form MFM-300($V^{IV}$), which were analysed by in situ single-crystal diffraction. MFM-300($V^{IV}$) retains the space group $I4_122$ and shows a slight contraction of the unit cell parameters (0.6% for a axis; 1.2% for c axis) with a similar channel size (Table 1). The most significant structural change upon oxidation is that the V–O bond length involving the bridging oxygen reduces dramatically from 1.948(2) to 1.838(1) Å (Table 1), which is between the typical bond distances for $V^{III}$–O ($\sim 1.95$ Å) and $V^{IV}=O$ ($\sim 1.65$ Å)[21]. Detailed investigation of the crystal structure revealed no residual electron density close to

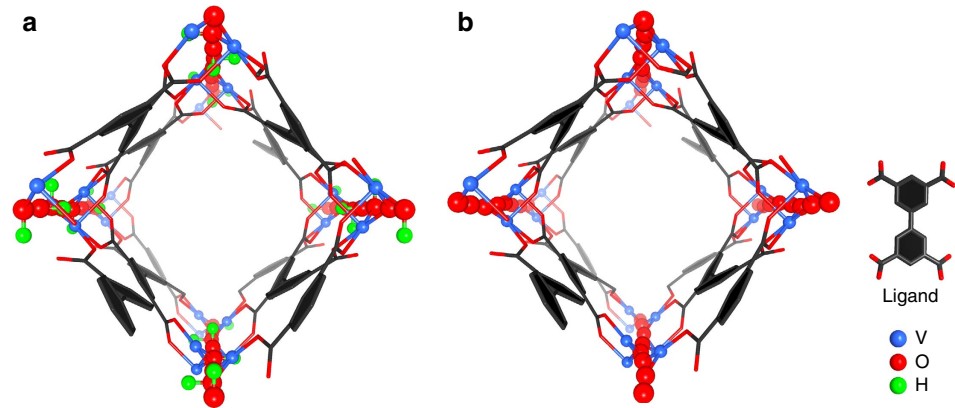

**Figure 1 | Structures of MFM-300(V).** Views of (**a**) MFM-300(V$^{III}$) and of (**b**) MFM-300(V$^{IV}$). The pore size is ~6.7 × 6.7 Å for both MFM-300(V$^{III}$) and MFM-300(V$^{IV}$) taking into consideration Van der Waals radii. The hydroxyl groups (green and red balls) in MFM-300(V$^{III}$) protrude into the channel, which change to O$^{2-}$ bridge (shown as red ball) in MFM-300(V$^{IV}$).

the bridging V$^{IV}$–O–V$^{IV}$ oxygen atom, indicating complete removal of the proton upon oxidation, which was further confirmed by neutron diffraction (Supplementary Fig. 2). Bond valence sum calculations give a valence of 3.03 for MFM-300(V$^{III}$) and 3.96 for MFM-300(V$^{IV}$). Quantitative oxidation of vanadium centres on going from MFM-300(V$^{III}$) to MFM-300(V$^{IV}$) has been further confirmed by XPS spectroscopy, which shows increased binding energy for the V$^{IV}$ species (Fig. 2 and Supplementary Fig. 3). IR spectroscopy confirms the disappearance in MFM-300(V$^{IV}$) of the $\nu_{O-H}$ stretching mode of the hydroxyl group at 3,639 cm$^{-1}$ observed in MFM-300(V$^{III}$) (Fig. 6a and Supplementary Fig. 4). Thus, the post-synthetic oxidation leads to retention of the porous structures in these two materials, and therefore affords an excellent platform to compare the supramolecular binding of guest molecules as a function of the presence and absence of the hydroxyl protons within the channel. The oxidized material MFM-300(V$^{IV}$) can be reduced back to MFM-300(V$^{III}$) using Na$_2$SO$_3$ in aqueous solution, with the colour of MFM-300(V$^{IV}$) changing from dark purple back to pale green, corresponding to the re-formation of MFM-300(V$^{III}$). PXRD data, however, confirm that some partial decomposition of the framework takes place under these conditions. Therefore, a quantitative reduction of MFM-300(V$^{IV}$) back to MFM-300(V$^{III}$) has not yet been achieved.

It should be noted that the post-synthetic oxidation of MFM-300(V$^{III}$) is currently the exclusive way of obtaining MFM-300(V$^{IV}$); hydrothermal syntheses using VOSO$_4$ or VO$_2$ salts based upon conditions reported[25] for other vanadium-based MOFs failed thus far to produce MFM-300(V$^{IV}$) in our hands. Interestingly, MIL-47(V$^{III}$) and MIL-47(V$^{IV}$) also show one-dimensional channels constructed from [V$_2$(OH)$_2$O$_4$]$_\infty$ chains bridged by terephthalate ligands[20]. MIL-47(V$^{III}$) and MIL-47(V$^{IV}$) can be synthesized directly from V$^{III}$ and V$^{IV}$ salts, respectively, but with considerable uncoordinated terephthalic acid encapsulated in the materials. Calcination of the as-synthesized complexes in air at 250 °C can effectively remove the free ligands to generate the corresponding activated materials. However, the V$^{III}$ centre was inevitably oxidized to V$^{IV}$ during calcination and the open form of MIL-47(V$^{III}$) could not be isolated until very recently when a new activation method based upon DMF extraction was developed that stabilized MIL-47(V$^{III}$) material. CO$_2$ and H$_2$O adsorption isotherms confirm that MIL-47(V$^{III}$) and MIL-47(V$^{IV}$) exhibit significantly different degrees of framework flexibility upon guest inclusion, and therefore

**Table 1 | Unit cell parameters, bond lengths and bond valence calculations of MFM-300(V$^{III}$) and MFM-300(V$^{IV}$).**

|  | MFM-300(V$^{III}$) | MFM-300(V$^{IV}$) |
|---|---|---|
| Formula | V$_2$(OH)$_2$(C$_{16}$H$_6$O$_8$) | V$_2$(O)$_2$(C$_{16}$H$_6$O$_8$) |
| Mr | 462.1 | 460.1 |
| Space group | $I4_1 22$ | $I4_1 22$ |
| $a, b$ (Å) | 15.117(1) | 15.024(1) |
| $c$ (Å) | 12.083(1) | 11.936(1) |
| Volume (Å$^3$) | 2761.4(5) | 2694.2(4) |
| V–O$_{bridging}$ (Å) | 1.978(1) | 1.838(1) |
| V–O$_{carboxylate}$ (Å) | 2.004(2) | 1.971(2) |
|  | 2.007(2) | 2.031(2) |
| ∠ V···O$_{bridging}$···V (°) | 125.6(1) | 134.7(2) |
| V···V distance (Å) | 3.519 | 3.392 |
| Bond valence calculation for vanadium | 3.027 | 3.960 |

cannot be directly compared in terms of host–guest chemistry[29]. In contrast, MFM-300(V) incorporates a wine-rack framework connection tied together by strong *cis*-$\mu_2$-OH groups, resulting in isostructural, highly rigid and robust frameworks in both V$^{III}$ and V$^{IV}$ materials.

Thermal gravimetric analysis (TGA) of MFM-300(V$^{III}$)-solv shows that water can be fully removed at 200 °C under N$_2$ flow, and no further weight loss was observed to 450 °C when decomposition sets in (Supplementary Fig. 5). In contrast, the TGA of MFM-300(V$^{IV}$) (prepared by heating under O$_2$) shows ~2% weight loss before 120 °C, attributed to adsorbed moisture during sample transportation for TGA measurement. No further weight loss was observed to 490 °C, where the MOF decomposed.

**Electron paramagnetic resonance analysis**. To probe the change in oxidation state of V centres, electron paramagnetic resonance (EPR) studies at X-band (9.86 GHz) and room temperature were undertaken on MFM-300(V$^{III}$) and MFM-300(V$^{IV}$) (Fig. 2). MFM-300(V$^{III}$) is EPR silent under these conditions, consistent with $d^2$ V$^{III}$ centre. In six-coordinate O$_h$ symmetry, the V$^{III}$, $d^2$ ion has a $^3T_{1g}$ ground state. In lower symmetry, a $^3$A state often results and spin–orbit coupling and gives rise to very large zero-field splitting (up to tens of cm$^{-1}$) of the $S = 1$ state (e.g., 5 cm$^{-1}$ in [V(H$_2$O)$_6$]$^{3+}$)[30]. Hence, most V$^{III}$ species are EPR silent at

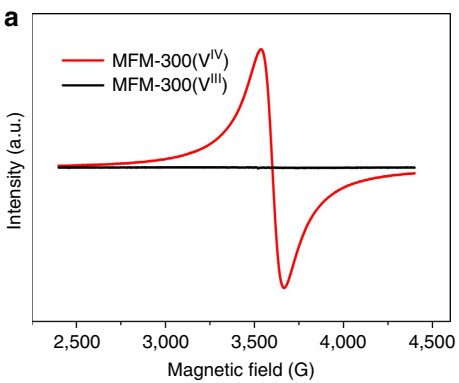

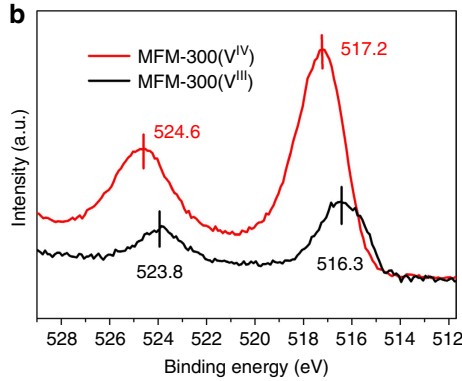

**Figure 2 | Electron paramagnetic (EPR) resonance and X-ray photoelectron spectra (XPS).** (**a**) Electron paramagnetic resonance (EPR) spectra and (**b**) X-ray photoelectron spectroscopy (XPS) of desolvated MFM-300($V^{III}$) and MFM-300($V^{IV}$). The X-band (9.86 GHz) EPR spectra were collected at 293 K. MFM-300($V^{III}$) is EPR silent under these conditions, consistent with $V^{III}$. MFM-300($V^{IV}$) has a strong EPR spectrum under the same conditions, giving a single resonance centred at $g = 1.955$ (9.8661 GHz), consistent with $V^{IV}$, $d^1$ ($g < g_e$, the free electron value), with a peak-to-peak linewidth of ca. 130 G.

X-band where the microwave energy ($hv$) is only ca. 0.3 cm$^{-1}$. In contrast, MFM-300($V^{IV}$) shows a strong EPR signal under the same conditions, giving a single resonance centred at $g = 1.955$, consistent with $V^{IV}$, $d^1$ ($g < g_e$, the free electron value), with a peak-to-peak linewidth of ca. 130 G. This broad linewidth is due to the exchange interactions in the $V^{IV}$ chains and results in the absence of any $^{51}V$ hyperfine resolution. Thus, EPR analysis confirms the change of oxidation state of $V^{III}$ to $V^{IV}$ in this system.

**Gas adsorption analysis.** $N_2$ isotherms at 77 K for both MFM-300(V) samples show no significant adsorption uptake, probably due to an activation diffusion effect at 77 K caused by the narrow pore channels. In contrast, low-pressure $CO_2$ isotherms of MFM-300($V^{III}$) at ambient temperatures (273–303 K) show very high uptake capacities with a value of 8.6 mmol g$^{-1}$ (37.8 wt%) recorded at 273 K and 1.0 bar (Fig. 4 and Supplementary Figs 6 and 7), surpassing those of other MFM-300 analogues (7.0 mmol g$^{-1}$ for MFM-300(Al)[26], 2.86 mmol g$^{-1}$ for MFM-300(Ga)[27] and 6.3 mmol g$^{-1}$ for MFM-300(In)[28]) under the same conditions. The $CO_2$ uptake of MFM-300($V^{III}$) at 298 K and 1.0 bar is 6.0 mmol g$^{-1}$ (26.4 wt%), comparable to the best-behaving materials such as MOF-74(Mg) or CPO-27(Mg) (26~27.5 wt%) under the same conditions[31]. The $CO_2$ uptake in MFM-300($V^{IV}$) is lower than that of its $V^{III}$ counterpart as a result of removal of the hydroxyl group from the pore. At 1.0 bar, the $CO_2$ uptake of MFM-300($V^{IV}$) is 3.54 mmol g$^{-1}$ (15.6 wt%) at 298 K and 6.56 mmol g$^{-1}$ (28.9 wt%) at 273 K, a reduction of 41 and 25%, respectively, in comparison with MFM-300($V^{III}$). Larger reductions in $CO_2$ uptake are observed at higher temperatures, consistent with reduced MOF-$CO_2$ binding strengths. The difference in $CO_2$ uptakes between these two materials is less pronounced with increasing pressure at a given temperature because the $CO_2$ intermolecular dipole interaction becomes more apparent at higher loadings. For example, at 273 K and 20 bar, the $CO_2$ uptake of MFM-300($V^{III}$) and MFM-300($V^{IV}$) were recorded to be 11.7 and 10.0 mmol g$^{-1}$, respectively, a difference of 14.5% (Fig. 4d). Analysis of the high-pressure $CO_2$ adsorption isotherm at 273 K by DFT/Monte-Carlo methods gives a surface area of 1,892 and 1,565 m$^2$ g$^{-1}$, a pore size distribution centred at 5.2 and 5.4 Å, and a cumulative pore volume of 0.490 and 0.481 cm$^3$ g$^{-1}$ for MFM-300($V^{III}$) and MFM-300($V^{IV}$), respectively (Fig. 4c and Supplementary Figs 8–13), confirming the microporous nature of

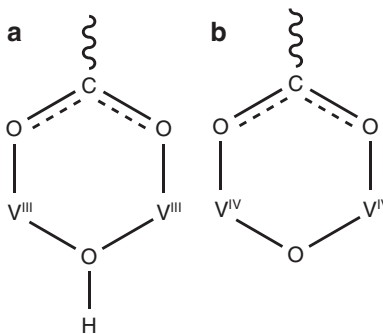

**Figure 3 | Coordination in MFM-300($V^{III}$) and MFM-300($V^{IV}$).** View of the bonding at the V centre in (**a**) MFM-300($V^{III}$) (**b**) and MFM-300($V^{IV}$).

these two materials. Interestingly, the slightly increased pore size (and wider distribution) of MFM-300($V^{IV}$) is consistent with the deprotonation of the hydroxyl group in the channel, leading to decreased $CO_2$ uptake at low pressure. The pore size distribution analysis confirms a slightly smaller pore size than that measured from the X-ray single crystal structure consistent with the presence of activated diffusion in the gas adsorption isotherms. The narrow pore size can also afford enhanced overlapping potential to gas molecules, increasing the overall host–guest binding interactions, and the slightly reduced pore volume of MFM-300($V^{IV}$) is responsible for its lower uptake capacity at high pressure in comparison with MFM-300($V^{III}$).

The isosteric heat of adsorption ($Q_{st}$) for $CO_2$ adsorption in MFM-300($V^{III}$) (obtained using van't Hoff Equation) lies in the range 27.5–28 kJ mol$^{-1}$ for uptakes of 0.5–1.0 mmol g$^{-1}$ and increases continuously thereafter to 32 kJ mol$^{-1}$ at 7 mmol g$^{-1}$, which is slightly higher than that of MFM-300(Al) by ~1.0 kJ mol$^{-1}$ (Fig. 4b). In comparison, the $Q_{st}$ for MFM-300($V^{IV}$) is 24 kJ mol$^{-1}$ at low $CO_2$ coverage and increases steadily to 29 kJ mol$^{-1}$ at 7 mmol g$^{-1}$, which is some ~3 kJ mol$^{-1}$ lower than that of MFM-300($V^{III}$), confirming the presence of a weaker adsorbate–adsorbent binding interaction in MFM-300($V^{IV}$). Nevertheless, the increasing trends of the variation of $Q_{st}$ plots are the same for these two materials, suggesting the presence of strong $CO_2$–$CO_2$ interactions with increasing surface coverage in both materials.

The binding energy of $CO_2$ in both MFM-300(V) materials has also been estimated by DFT calculations at a loading of

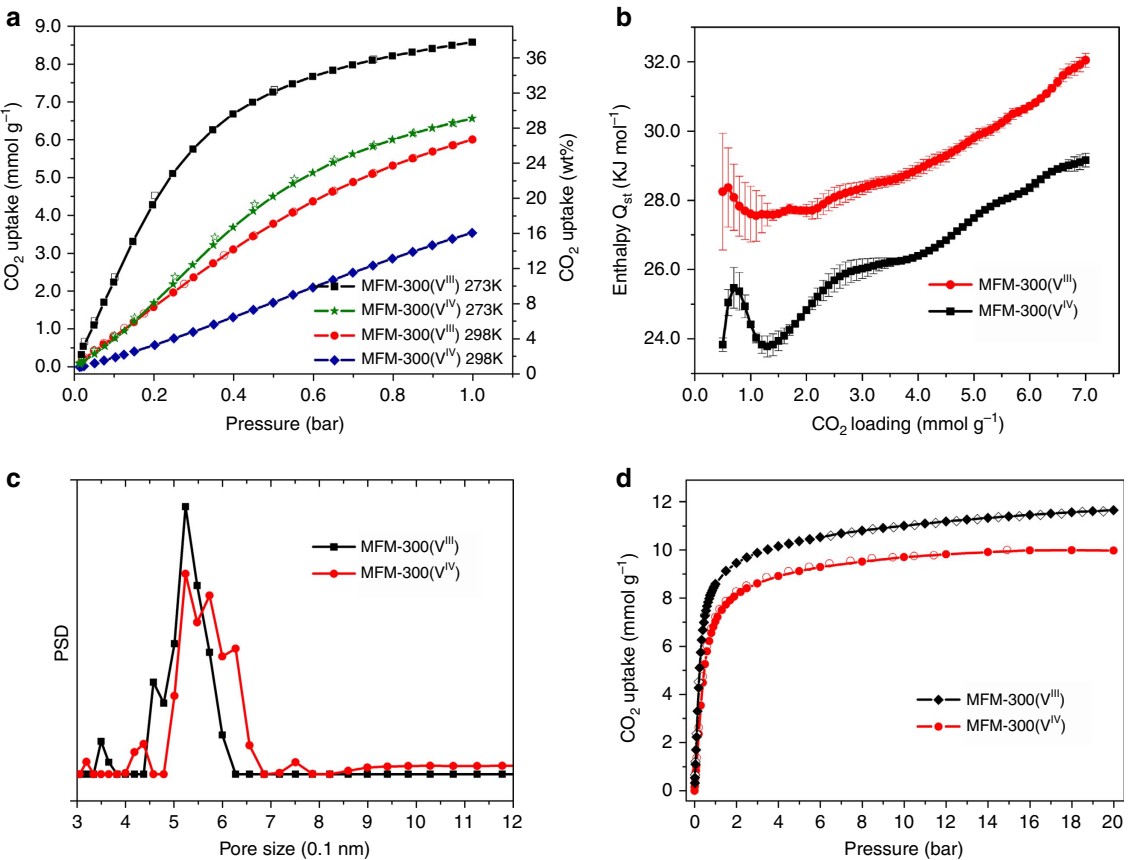

**Figure 4 | View of the gas adsorption data.** (**a**) Comparison of the $CO_2$ adsorption isotherms for MFM-300($V^{III}$) and MFM-300($V^{IV}$) at 273 and 298 K and 1.0 bar. (**b**) Variation of isosteric heat of adsorption $Q_{st}$ as a function of $CO_2$ uptake for MFM-300($V^{III}$) and MFM-300($V^{IV}$) calculated using the van't Hoff isochore on $CO_2$ adsorption isotherms measured at 273–303 K. Standard deviations are represented by error bars. (**c**) Comparison of the pore size distribution plots for MFM-300($V^{III}$) and MFM-300($V^{IV}$), suggesting a slightly wider pore size in the latter material. (**d**) Comparison of the $CO_2$ adsorption isotherms for MFM-300($V^{III}$) and MFM-300($V^{IV}$) at 273 K up to 20 bar.

8.6 mmol g$^{-1}$ (equivalent to 2 $CO_2$/V). The stepwise calculation was carried out by (i) optimizing the structure of the bare MOF by finding the local potential energy minimum with the final potential energy $E_1$; (ii) optimizing the structure of the $CO_2$-loaded MOF (2 $CO_2$/V) with the final potential energy $E_2$; (iii) removing the MOF host and leaving the $CO_2$ molecules in the system to minimize the energy $E_3$. The $CO_2$ binding energy ($\Delta E$), obtained by the calculation of $\Delta E = E_1 - E_2 + E_3$, was 34.9 and 29.7 kJ mol$^{-1}$ for MFM-300($V^{III}$) and MFM-300($V^{IV}$), respectively, consistent with the $Q_{st}$ analysis and confirming a lower $CO_2$-MOF binding strength in MFM-300($V^{IV}$).

**Structural analysis by neutron powder diffraction.** In order to determine the binding sites of $CO_2$ within these two materials, neutron powder diffraction (NPD) experiments were carried out for $CO_2$-loaded MFM-300($V^{III}$) and MFM-300($V^{IV}$) samples at 7 K. NPD data enabled full structural analysis via Rietveld refinement (Supplementary Figs 14–17 and Supplementary Table 2), yielding the positions, orientations and occupancies of adsorbed $CO_2$ molecules within the materials. $CO_2$-loaded MFM-300($V^{III}$) retains the space group $I4_122$ and two crystallographically independent $CO_2$ sites (I and II) were observed (Fig. 5). Molecules $CO_2^I$ are disordered with the respect to a twofold rotation axis and are in an end-on mode interacting with the hydroxyl group with a OH$\cdots$O$_{CO_2}$ distance of 1.863(1) Å (Fig. 5 and Supplementary Fig. 18), shorter than that obtained for MFM-300(Al) (2.298(4) Å) by PXRD at 273 K (ref. 20). This suggests a

stronger hydrogen bond between adsorbed $CO_2$ and the HO–$V^{III}$ group and probably reflects the difference in the acidity between the bridging Al–OH and V–OH groups. $CO_2^{II}$ binds to the carboxylate oxygen atoms via a dipole interaction with $C_{CO_2}\cdots O_{carboxylate}$ distances ranging from 2.984(2) to 3.369(2) Å. Further dipole interactions were observed between adsorbed $CO_2$ molecules on sites I and II with intermolecular $C^I\cdots O^{II}$ distance of 3.103(1) Å in a typical T-shape arrangement (Fig. 5c), comparable to that observed in dry ice (3.178(1) Å)[32]. Supplementary intermolecular $\pi\cdots\pi$ interactions are also observed between the $CO_2$ molecules at site II with a $C^{II}\cdots O^{II}$ distance of 2.565(2) Å.

The site occupancy of adsorbed $CO_2$ molecules was obtained by free refinement against the NPD data (Table 2). At a $CO_2$ loading of 1.4 $CO_2$/V, the occupancies of $CO_2^I$ and $CO_2^{II}$ are found to be 0.82 and 0.29, respectively. As the $CO_2$ loading was increased to 2.0 $CO_2$ per V, the occupancy of $CO_2^I$ increased slightly to 0.99, suggesting saturation of site I, while the occupancy of $CO_2^{II}$ almost doubled from 0.29 to 0.52. This result indicates that site I which contacts with the hydroxyl group directly is the preferred location for adsorbed $CO_2$ molecules at low surface coverage. The $CO_2$ molecules occupying site I have a sequential effect on the population of site II, modulated by the intermolecular dipole interaction.

In contrast, the NPD pattern of $CO_2$-loaded MFM-300($V^{IV}$) shows a change in the space group from $I4_122$ to $P4_122$ as indicated by the appearance of Bragg peaks ($hkl$, $h+k+l = 2n+1$), which are systematically absent in $I4_122$ (Supplementary

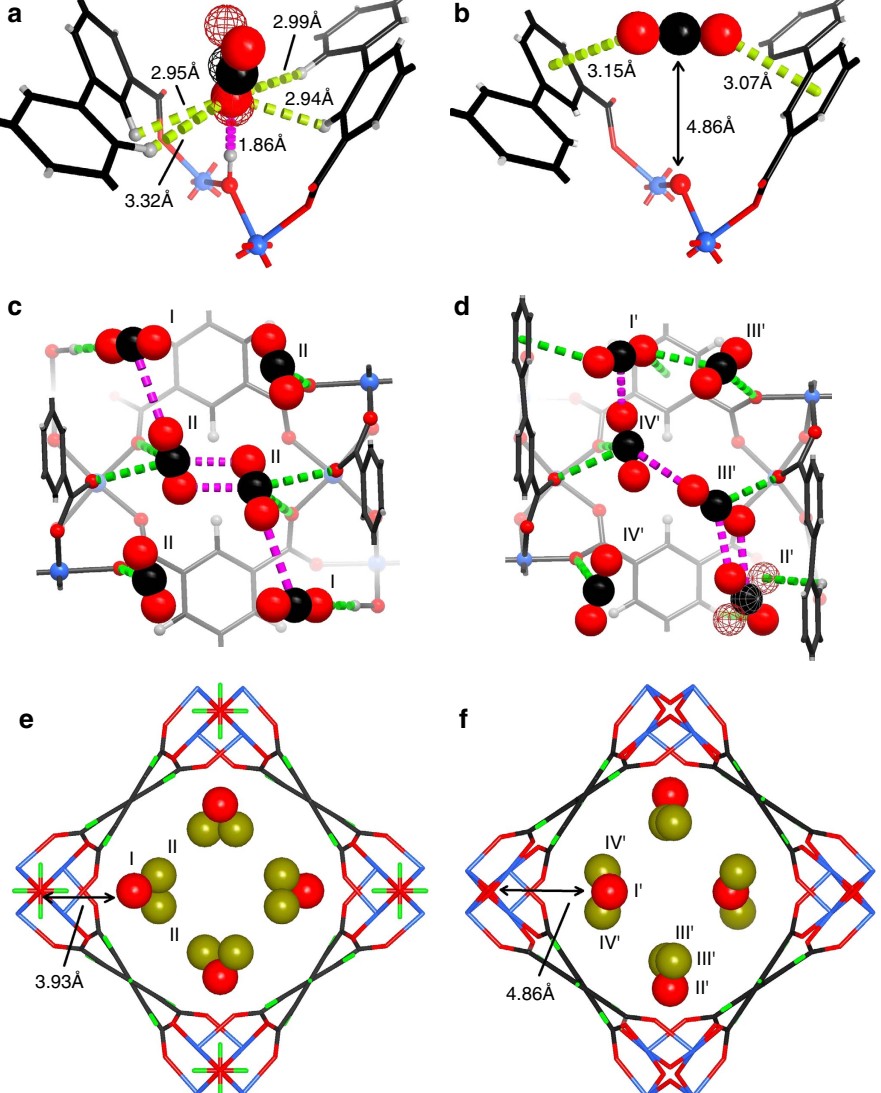

**Figure 5 | Comparison of the crystal structures for $CO_2$-loaded MFM-300($V^{III}$) and MFM-300($V^{IV}$) studied by NPD at 7 K.** (**a**) Interactions of $CO_2$ molecule at site I with the hydroxyl and the –CH on phenyl groups in MFM-300($V^{III}$). (**b**) 'Sandwiched' binding mode between $CO_2$ molecule on site I' with the phenyl groups in MFM-300($V^{IV}$). (**c**) $CO_2$ molecules at sites I and II within MFM-300($V^{III}$) (disordered $CO_2$ on site I is shown in the averaged positions). The $C_{CO_2} \cdots O_{carboxylate}$ distances are in the range of 2.984(2) to 3.369(2) Å. The C$\cdots$O distance between T-shape arranged $CO_2$ is 3.178(1) Å and that between π–π interacted $CO_2$ is 2.565(2) Å. (**d**) $CO_2$ molecules at sites I' to IV' within MFM-300($V^{IV}$). The $H_{phenyl} \cdots O_{CO_2}$ distance is 2.666(2) Å. The $C_{CO_2} \cdots O_{carboxylate}$ distances are in the range of 2.843(2) to 3.057(2) Å. C$\cdots$O distances between T-shape arranged $CO_2$ are in the range of 2.478(1) to 3.132(1) Å and that between π–π interacted $CO_2$ is 2.741(2) Å (carbon, black; hydrogen, light grey; oxygen, red; vanadium, light blue; interactions between $CO_2$ and frameworks are shown in green dashed line; interactions between adsorbed $CO_2$ molecules are shown in pink dashed line). (**e**) Binding sites for adsorbed $CO_2$ within MFM-300($V^{III}$). (**f**) Binding sites for $CO_2$ within MFM-300($V^{IV}$) (red and dark yellow balls indicate the centre of the corresponding binding site).

**Table 2 | Occupancy of adsorbed $CO_2$ molecules at different loading from the refinement against NPD data*.**

| Site | $CO_2$ loading | I (I') | II (II') | III' | IV' |
|---|---|---|---|---|---|
| MFM-300($V^{III}$) | 1.4 | 0.824(3) | 0.584(3) | — | — |
| | 2.0 | 0.992(3) | 1.046(3) | — | — |
| MFM-300($V^{IV}$) | 1.4 | 0.277(7) | 0.169(7) | 0.430(6) | 0.497(6) |
| | 1.6 | 0.297(8) | 0.179(8) | 0.538(6) | 0.577(6) |

*Loadings and occupancies are normalized to the metal ion, and are presented as $CO_2$ molecule per V centre.

Fig. 19). Nevertheless, the framework skeleton remains the same as that of MFM-300($V^{III}$) and of the single crystal structure of bare MFM-300($V^{IV}$) with four independent binding

sites (I', II', III' and IV') for adsorbed $CO_2$ molecules being determined for $CO_2$-loaded MFM-300($V^{IV}$). Most notably, the $CO_2$ molecule at site I' within MFM-300($V^{IV}$) is rotated by

90° (compared with that in MFM-300($V^{III}$)) with respect to the bridging $O^{2-}$ group and is thus side-on to the $O^{2-}$ centre. It is thus sandwiched within a cleft between two phenyl rings of adjacent ligands, resulting in a highly translationally hindered site owing to the geometrical confinement (Fig. 5b). The linear $CO_2$ points to the plane of two phenyl rings with an $O_{CO_2} \cdots$ c.g.$_{phenyl}$ distance of 3.069(2) and 3.146(3) Å. To our best knowledge, $CO_2$ binding between two phenyl rings in a 'sandwich' mode has not previously been experimentally observed. The $CO_2$ molecule at site II′ is disordered over two orientations, forming contacts to the aromatic hydrogen atoms ($H_{phenyl} \cdots O_{CO_2} = 2.666(2)$ Å). $CO_2$ molecules at sites III′ and IV′ are close to the carboxylate group of the ligand with the $C_{CO_2} \cdots O_{carboxylate}$ distance ranging from 2.843(2) to 3.057(2) Å (Fig. 5d). In the absence of strong interactions to the MOF host, adsorbed $CO_2$ molecules in MFM-300($V^{IV}$) interact between each other with short $C \cdots O$ distances in the range of 2.478(1) to 3.132(1) Å, and the overall packing of $CO_2$ in MFM-300($V^{IV}$) is more close to the dry ice structure compared with that in MFM-300($V^{III}$), suggesting the presence of stronger intermolecular dipole interactions in MFM-300($V^{IV}$) (Supplementary Fig. 20). Refinement of the site occupancies suggests that 20 and 10% of adsorbed $CO_2$ resides at site I′ and II′, respectively, at a loading of 1.4 $CO_2$ per V, much lower than that (60%) for the corresponding position (site I) within MFM-300($V^{III}$) at the same $CO_2$ loading. This leads to 70% of adsorbed $CO_2$ molecules residing on sites III′ and IV′, showing a dipole interaction with the carboxylate oxygen centres, clearly suggesting that these sites (III′ and IV′) now become the primary binding sites in MFM-300($V^{IV}$) in the absence of the hydroxyl proton.

The location of adsorbed $CO_2$ molecules have been determined in a number of MOF systems that primarily contain open metal sites (e.g., HKUST-1 (ref. 33) and MOF-74 (ref. 23)) or pendent functional groups (e.g., amine group[24] and hydroxyl group[26]). These studies suggest that the open metal sites or functional groups in the pore have a specific role in $CO_2$ binding to the localized environment; their effects upon $CO_2$ binding on a macroscopic level are however generally overlooked. The present study of MFM-300($V^{III/IV}$) offers a comprehensive understanding of the effect of functional groups in $CO_2$ binding. It is found that the hydroxyl proton plays a significant role in $CO_2$ binding, not only to its localized environment but also to those $CO_2$ molecules residing in the centre of the pore. Thus, the overall arrangement and packing of $CO_2$ molecules in the extended channel of these two materials are distinct with different site populations [60/40 for Site I/II in MFM-300($V^{III}$) vs 30/70 for Sites I′ + II′/III′ + IV′ in MFM-300($V^{IV}$)].

**Structural analysis by DFT calculation.** The structures for $CO_2$-loaded MFM-300($V^{III}$) and MFM-300($V^{IV}$) have also been optimized by DFT calculation at the loading of 2$CO_2$ per V. The optimized structural model of MFM-300($V^{III}$) $\cdot$ 4$CO_2$ shows excellent agreement with that obtained from NPD experiment, confirming the end-on model of adsorbed $CO_2$ molecules interacting to the hydroxyl groups ($H \cdots O_{CO_2} = 2.076$ Å, Fig. 5e and Supplementary Fig. 21a). In contrast, for MFM-300($V^{IV}$), the calculated structure model does not show the 'sandwich' mode of adsorbed $CO_2$ between two phenyl rings as seen in the NPD model. This is probably because that the $CO_2$-loaded MFM-300($V^{IV}$) system has a less distinct energy minimum than the $V^{III}$ material, that is, the rotation/translation of adsorbed $CO_2$ molecules can occur more readily in MFM-300($V^{IV}$), leading to a series of similar structural models with close energy minimum in the calculation. That

said, the change of the overall arrangement and packing of $CO_2$ molecules in the pore has been successfully predicted by DFT calculations on going from MFM-300($V^{III}$) to MFM-300($V^{IV}$), consistent with the presence of weaker $CO_2$–host and stronger $CO_2$–$CO_2$ binding interaction in MFM-300($V^{IV}$) (Fig. 5f and Supplementary Fig. 21b).

**Analysis of $CO_2$ binding via inelastic neutron scattering.** Understanding the change in dynamics upon adsorption of $CO_2$ within these porous hosts can provide fundamental insight into the formation of multiple supramolecular contacts (e.g., hydrogen bond and intermolecular dipole interactions) with the adsorbed gas molecules. INS is a powerful neutron spectroscopic technique that is particularly sensitive to the dynamics of hydrogen atoms owing to the exceptionally large cross-section for that isotope and is therefore used here to investigate the vibrational states of hydrogen bonded systems. Comparison of the INS spectra of the bare MFM-300($V^{III}$) and MFM-300($V^{IV}$) samples shows a number of differences (Fig. 6 and Supplementary Fig. 22). The most significant change is the absence of the –OH out-of-plane bending mode at 47 meV and disappearance of a series of hydroxyl deformation modes between 40 and 70 meV in MFM-300($V^{IV}$), confirming the deprotonation of the bridging hydroxyl group upon oxidation. The peaks corresponding to the C–H wagging/bending modes (80–167 meV) remain the same between these two samples. Calculated INS spectra for these two MOFs by DFT were found to be in good agreement with the experimental INS data (Supplementary Figs 23 and 24). On addition of $CO_2$ in MFM-300($V^{III}$), all peaks shift slightly to higher energy, indicating a stiffening effect of the host–guest lattice. Importantly, increases in intensity were observed for the peaks at 48 and 118 meV, indicating the change of the vibrational motion of both –OH and –CH groups upon $CO_2$ binding, consistent with the formation of hydrogen bonds and supramolecular interactions as observed in the NPD and DFT models (Fig. 6c). This is further confirmed by the difference spectrum, and significantly these changes are in excellent agreement with the calculated difference INS spectrum (Fig. 6d). In contrast, addition of $CO_2$ in MFM-300($V^{IV}$) is accompanied by changes of the peak intensity at 32, 85, 117 and 158 meV, corresponding to the distortion of the MOF lattice, distortion of the phenyl rings, deformation of the phenyl ring and in-plane C–H bending modes, respectively. This result confirms unambiguously that the phenyl ring and –CH group now become the primary binding site for adsorbed $CO_2$ in MFM-300($V^{IV}$), in excellent agreement with the NPD results. These changes of the INS peaks are successfully predicted from the DFT calculations with an exceptionally high degree of consistency (Fig. 6f). Thus, the INS results have confirmed (i) the deprotonation of the bridging hydroxyl group occurs on going from MFM-300($V^{III}$) to MFM-300($V^{IV}$); (ii) in MFM-300($V^{III}$), both –OH and –CH are active binding sites for adsorbed $CO_2$ via formation of hydrogen bonds; (iii) in MFM-300($V^{IV}$), the –CH and phenyl groups are the active binding sites for adsorbed $CO_2$ via formation of supramolecular contacts. These observations are consistent with the NPD/DFT studies and confirm the significant effects of bridging hydroxyl groups on guest binding in MFM-300($V^{III}$).

**Discussion**

It is widely accepted that introduction of pendent functional groups into the pores of MOFs offers accessible binding sites for guest molecules by attracting them via specific interactions. These sites often lose activity after reaching a 1:1 host–guest complex and are thus believed to have a limited role in the entire

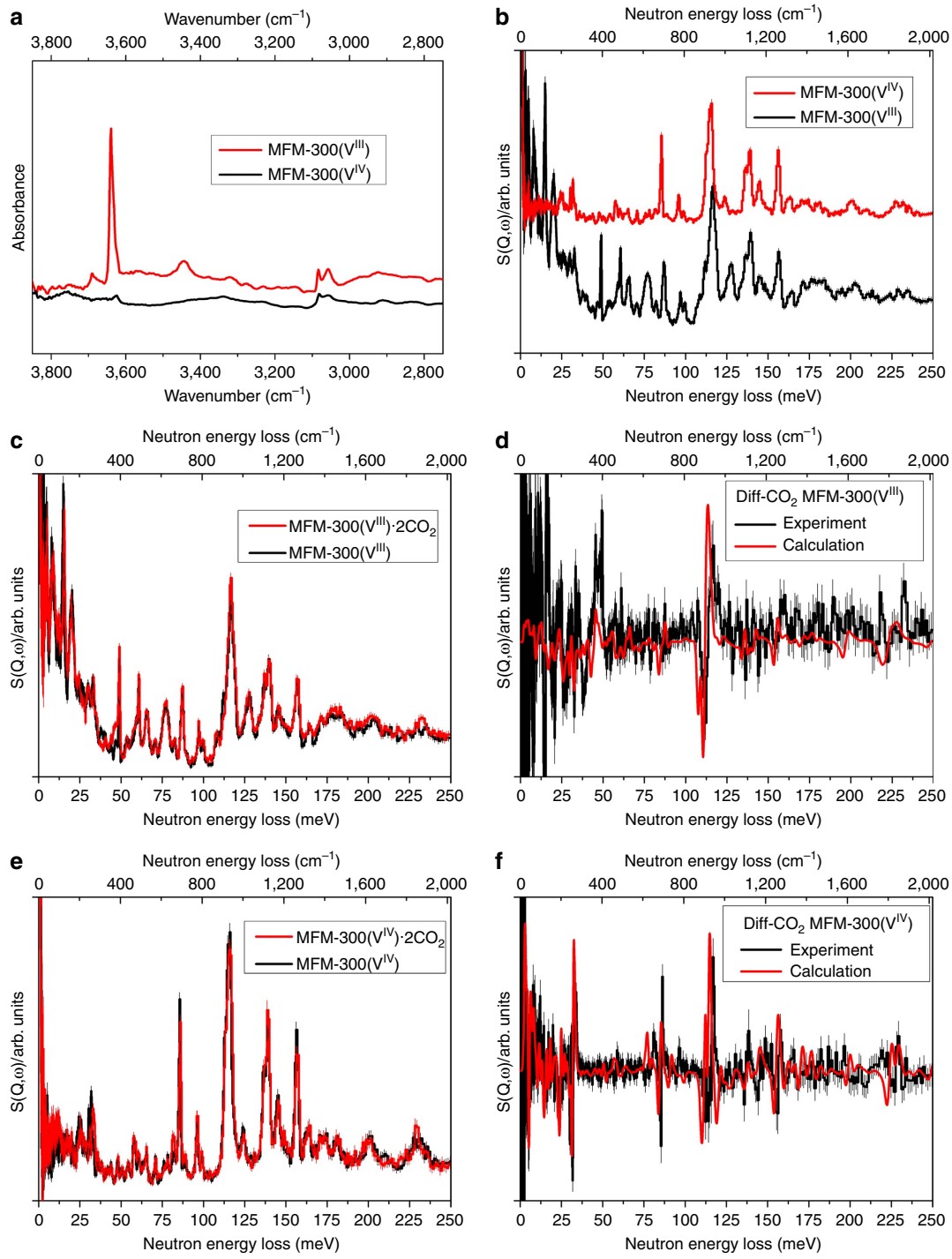

**Figure 6 | Synchrotron micro-IR and inelastic neutron scattering (INS) spectra for MFM-300(V).** Comparison of (**a**) the synchrotron micro-IR spectra and (**b**) INS spectra for bare MFM-300(V$^{III}$) and MFM-300(V$^{IV}$). IR and INS spectra offer vibrational information on the –OH stretching and deformation region, respectively, and are thus complementary. (**c**) Comparison of the INS spectra for bare and $CO_2$-loaded MFM-300(V$^{III}$); (**d**) difference spectra for both experiment and calculation showing changes of the H motion from the local MOF upon $CO_2$ binding in MFM-300(V$^{III}$). (**e**) Comparison of the INS spectra for bare and $CO_2$-loaded MFM-300(V$^{IV}$); (**f**) difference spectra for both experiment and calculation showing changes of the H motion from the local MOF upon $CO_2$ binding in MFM-300(V$^{IV}$). Synchrotron IR and INS data were collected at 298 and 10 K, respectively.

gas uptake process. Many functional groups contain hydrogen donors (e.g., –OH, –NH$_2$, –OCNH–, COOH, –CH$_3$), and the direct visualization of the molecular details for host–guest binding involving potential formation of hydrogen bonds is of fundamental importance to understand the function of host–guest systems. We have reported here the synthesis and

characterization of a porous material, MFM-300(V$^{III}$), and its single-crystal to single-crystal transformation via post-oxidation to give its counterpart MFM-300(V$^{IV}$). The oxidation of V centres from +3 to +4 promotes deprotonation of the bridging hydroxyl group, achieving fine tuning of the pore environment. It has been confirmed that these protons play a key

role in the $CO_2$ adsorption (both uptakes and heats of adsorption) in adsorption experiments. A study combining NPD, INS and modelling has unambiguously determined the binding sites and structural dynamics for these $CO_2$–host systems. We have confirmed that the proton on the hydroxyl group can not only attract and localize adsorbed $CO_2$ molecules via direct formation of hydrogen bonds but also affects the macroscopic packing and arrangement of $CO_2$ molecules in the extended channel. Interestingly, a form of $CO_2$ binding to the phenyl rings has been observed in MFM-300($V^{IV}$) by the NPD study. Thus, this report has provided direct evidence of the role of protons for guest binding in porous MOFs and, more importantly, it suggests that the effect of the functional group (i.e., protons) goes beyond specific binding to the local site and extends to the entire pore structure.

## Methods

**Synthesis of single crystals of MFM-300($V^{III}$)-solv.** A mixture of $VCl_3$ (0.076 g, 0.48 mmol), deionized water (5.0 ml) (degassed previously under Ar for 0.5 h), biphenyl-3,3′,5,5′-tetracarboxylic acid (0.010 g, 0.03 mmol) and hydrochloric acid (1.6 M, 1.0 ml) was transferred into a 23 ml Teflon lined autoclave, which was then charged with argon and sealed. Reaction took place at 210 °C for 3 days and afforded dark green prism-shaped crystals accompanied with colourless block-shaped crystals (which are MFM-300($V^{III}$)-solv and the re-crystallized ligand, respectively). The product was soaked in fresh $N,N'$-dimethylformamide (DMF) overnight to remove re-crystallized ligand, pure MFM-300($V^{III}$)-solv single crystals (ca. 5% yield) were collected by filtration and preserved in acetone.

**Synthesis of bulk MFM-300($V^{III}$)-solv.** A mixture of $VCl_3$ (0.40 g, 2.55 mmol) in deionized water (10.0 ml) (degassed previously under Ar for 0.5 h), biphenyl-3,3′,5,5′-tetracarboxylic acid (0.14 g, 0.42 mmol) and hydrochloric acid (0.3 M, 2.0 ml) was transferred into a 45-ml Teflon lined autoclave, which was then charged with Ar and sealed. The reaction was heated at 210 °C for 3 days to produce MFM-300($V^{III}$)-solv as a green microcrystalline powder (90% yield). This was soaked in fresh DMF overnight, and collected by filtration and stored under acetone.

**Preparation of desolvated MFM-300($V^{III}$).** MFM-300($V^{III}$)-solv was placed in a Schlenk flask, which was evacuated to $< 1 \times 10^{-3}$ mbar at ambient temperature for 1 h, and then heated at 150 °C for overnight under vacuum. Activated MFM-300($V^{III}$) was obtained as green powder and stored in a glovebox. An important issue in desolvating MFM-300($V^{III}$) is to avoid oxidation. Elemental analysis for MFM-300($V^{III}$), $C_{16}H_8O_{10}V_2$ (found (calculated%)): C 41.4(41.6), H 1.9(1.7), N 0.0(0.0).

**Post-synthetic oxidation of MFM-300($V^{III}$) to prepare MFM-300($V^{IV}$).** Single crystals or powder MFM-300($V^{III}$) was heated under pure $O_2$ flow at 150 °C overnight using a tube furnace. The temperature ramping speed was set to 1.0 °C min$^{-1}$. MFM-300($V^{IV}$) was obtained as dark purple solid that was stored under $N_2$ in a glovebox. Elemental analysis for MFM-300($V^{IV}$), $C_{16}H_6O_{10}V_2$ (found (calculated)%): C 41.5(41.8), H 1.5(1.3), N 0.0(0.0).

***In situ* single-crystal diffraction and structure determination.** A single crystal of MFM-300($V^{III}$) was selected and glued onto a MiTeGen loop with Loctite Double Bubble epoxy. This was inserted into the holder of the Advanced Light Source gas environmental cell at Beamline 11.3.1, comprising a 50 µm quartz glass tube coupled to gas handling and vacuum equipment. The cell was attached to a Bruker D8 diffractometer with a PHOTON100 CMOS detector on Beamline 11.3.1 at the Advanced Light Source equipped with an Oxford Cryosystems Cryostream 800 plus. Spheres of data were collected with $\lambda = 0.7749$ Å radiation, from a channel cut Silicon [111] monochromator, at various sample temperatures and pressures. The *in situ* oxidation was carried out in a flow-type gas cell via flowing air through the cell that was heated at 150 °C for 2 h (see Supplementary Method for more details of structure determination).

**EPR measurements.** X-band (9.86 GHz) EPR spectra were recorded on powder samples of MFM-300($V^{III}$) and MFM-300($V^{IV}$) at room temperature, using a BrukerEMX Micro EPR spectrometer equipped with an X-band microwave bridge, a high-Q resonator and a 1.0 T electromagnet. The microwave frequency was measured with a built-in digital counter and the magnetic field was calibrated using a Bruker strong pitch reference sample ($g = 2.0028$). Modulation amplitude and microwave power used were 3 G and 2 mW, respectively. Data were analysed with the Bruker Win EPR Simfonia package.

**Neutron powder diffraction.** NPD experiments were undertaken at the WISH diffractometer at the ISIS Facility. MFM-300(V) was loaded into a 6 mm diameter vanadium sample can and outgassed at $1 \times 10^{-7}$ mbar and 100 °C for 1 day. The sample was then loaded into a liquid helium cryostat and cooled to 7 K for data collection. $CO_2$ gas was introduced by warming the samples to 290 K, and the gas was dosed volumetrically from a calibrated volume. The gas-loaded sample was then cooled to 7 K over a period of 2 h to ensure good mobility of adsorbed $CO_2$ within the crystalline structure of MFM-300(V). The sample was kept at 7 K for an additional half an hour before data collection to ensure the thermal equilibrium (see Supplementary Method for more details).

**Inelastic neutron scattering.** INS experiments were undertaken using the TOSCA spectrometer at the ISIS Facility. MFM-300(V) was loaded into an 11 mm diameter vanadium sample can and outgassed at $1 \times 10^{-7}$ mbar and 100 °C for 1 day. The sample was loaded into a helium closed cycle refrigerator cryostat and cooled to 11 K for data collection. $CO_2$ gas was introduced by warming the samples to 290 K, and the gas was dosed volumetrically from a calibrated volume. The gas-loaded sample was then cooled to 11 K over a period of 2 h to ensure good mobility of adsorbed $CO_2$ within the crystalline structure of MFM-300(V). The sample was kept at 11 K for an additional half an hour before data collection to ensure the thermal equilibrium (see Supplementary Method for more details of DFT Calculation for INS spectra).

**Infrared measurements of gas-loaded MFM-300(V).** Infrared spectroscopic measurements of crystals of MFM-300($V^{III}$) and MFM-300($V^{IV}$) were carried out using a Bruker Hyperion 3000 microscope equipped with an $LN_2$ cooled MCT detector, coupled to a Bruker Vertex spectrometer supplied with broad band radiation from beamline B22 of the Diamond Light Source. Experiments were carried out using a Linkam FTIR600 environmental gas stage under a constant (100 cm$^3$ min$^{-1}$) flow of dry He. The desolvated sample was generated *in situ* by heating the sample to 393 K for 2 h, and 256 scan spectra were collected at 298 K.

**Data availability.** Single crystal data of MFM-300($V^{III}$) and MFM-300($V^{IV}$) are deposited at Cambridge Crystallographic Data Centre (CCDC) with numbers of 1479680–1479681, and numbers of 1481144–1481147 are for $CO_2$-loaded MFM-300(V) samples obtained from NPD. The data that support the findings of this study are available from the corresponding author on request.

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

## Acknowledgements

We thank EPSRC, ERC and University of Manchester for funding. We thank EPSRC for funding of the UK National EPR Facility at Manchester. We are especially grateful to STFC and the ISIS Neutron Facility for access to the Beamlines TOSCA and WISH, to Diamond Light Source for access to Beamline B22, to the Advanced Light Source for access to Beamline 11.3.1 and to ORNL for access to Beamline VISION. The computing resources were made available through the VirtuES (Virtual Experiments in Spectro-scopy) project, funded by Laboratory Directed Research and Development program (LDRD 7739) at the Oak Ridge National Laboratory. The Advanced Light Source is supported by the Director, Office of Science, Office of Basic Energy Sciences, of the U.S. Department of Energy under Contract No. DE-AC02-05CH11231. The develop-ment of the gas cell used in this work was partially funded by the Center for Gas Separations Relevant to Clean Energy Technologies, an Energy Frontier Research Center funded by the U.S. Department of Energy, Office of Science, Basic Energy Sciences under Award # DE-SC0001015.

## Author contributions

Z.L., H.G.W.G., T.L.E.: syntheses, characterization of MOF samples, measurements and analysis of adsorption isotherms. Z.L., S.J.T., K.J.G. and S.Y.: syntheses, collection and analysis of single crystal structures. Z.L., F.T., and E.J.L.M.: collection and analysis of the EPR data. M.Sa., T.L.E. and M.D.F.: collection of synchrotron IR data. Z.L., H.G.W.G., I.S., Y.C., M.Sa., P.M., S.R., A.J.R.-C, S.Y.: collection and analysis of neutron scattering data. S.Y. and M.Sc.: overall direction and design of project. Z.L., S.Y. and M.Sc. with all authors: preparation of the manuscript.
