## [Peer Review File · Nature Communications]

Reviewers' comments:

Reviewer #1 (Remarks to the Author):

Manuscript: Modulating Supramolecular Binding of Carbon Dioxide in a Redox-Active Porous Host

By Lu et al.

This is a very nice paper coming from highly recognized research groups. In the present paper, authors try to identify the adsorption sites for CO₂ in two iso-structural MOFs after a modification in the vanadium oxidation state (III or IV). Theoretical calculation predict pore channels in both samples around 0.67 x 0.67 nm (cross-section), large enough to accommodate N₂ and/or CO₂. However, N₂ adsorption measurements at 77K clearly anticipate that this small molecule (kinetic diameter 0.36 nm) is not able to access the porosity in both materials. Even N₂ adsorption measurements at 273 K are affected by these kinetic restrictions, preferentially in sample MFM-300 (VIII) (for instance, the N₂ adsorption isotherm at 273 K – black symbol in Figure S8- exhibit an upward deviation above 0.5 bar anticipating these kinetic limitations). The presence of these kinetic restrictions to access the inner porous structure can also affect a rather similar molecule such as CO₂ (0.33 nm kinetic diameter) as can be appreciated by comparing the low and high pressure CO₂ measurements. Whereas at atmospheric pressure sample MFM-300 (VIII) exhibits a larger adsorption capacity compared to sample MFM-300 (VIV), at high pressures (20 bar) both samples (III and IV) are rather similar, thus confirming the problems of accessibility at 1 bar. In summary, gas adsorption measurements clearly show that the pore size in these MOFs is smaller than theoretical predictions, these adsorption measurements being highly affected by kinetic restrictions.

As described above, CO₂ adsorption measurements at 273 K and 1 bar show clear differences depending on the vanadium oxidation state. Authors attribute this behaviour to the role of the –OH group in the supramolecular binding of CO₂, as predicted by DFT calculations. However, DFT calculation predict a complete occupation of these sites (CO₂I) at very low surface coverage, i.e. the effect of these hydroxyl groups in the CO₂ adsorption performance, if any, would be observed at very low relative pressures (e.g., p/p₀ < 10⁻⁵). Unfortunately, these values are far from the pressures measured in Figure 3a (highly above low coverages). The main experimental evidence about the role of the hydroxyl groups could come from INS. The comparison of the INS for samples III and IV (Figure 5b) clearly show important differences below 75 meV mainly due to the O-H vibration, torsion, bending and tunnelling modes. However, upon CO₂ adsorption this region is mainly unaffected (see for instance the difference spectra in Figure 5d), the main difference being at 110 meV due to the inhibited C-H vibrational motion.

In summary, although DFT calculation predict a promoting role of –OH groups in sample MFM-300 (VIII) for CO₂ adsorption, the presence of kinetic restrictions in the gas adsorption measurements, the lack of information at very low relative-pressures and the absence of a clear evidence in the INS measurements makes the final conclusion very speculative.

Reviewer #2 (Remarks to the Author):

Schroeder, Yang and coworkers report on the formation of a redox active V(III) metal organic framework containing bridging hydroxide ions which is transformed in its V(IV) counterpart upon exposition to O₂ at 150°C. The main point addressed in this report is the impact of the formation of hydrogen bonding interactions with quadrupolar carbon dioxide guest molecules. The results are indicative of a significant interaction of the CO₂ guest molecules with the hydroxide ions as deduced from end-on configuration of the CO₂ molecule with the hydroxide ion, as determined from neutron diffraction, and the high values of isosteric heat of adsorption. As shown in the

results the oxidation of the V(III) centers to V(IV) gives rise to the deprotonation of the hydroxide ion to an oxide ion with a concomitant modification in the CO₂ binding.

The results are of interest to the MOF community and deserves publication.

A few remarks:

Is the V(III) to V(IV) oxidation complete? Is there any possibility of V(III)/V(IV) mixed valence state?

Could this be confirmed by electronic spectroscopy and/or magnetic measurements?

Is the MOF oxidation reversible?

Minor point: the authors use the term hydroxy through out the text. Should not be hydroxide a more correct term?

Reviewer #3 (Remarks to the Author):

Below are responses according to the review criteria A-H. Overall, I recommend publication if the points below can be addressed.

A) In this work, the authors employ isostructural MFM-300(VIII) and (VIV) to explore, using neutron diffraction and inelastic scattering, how the supramolecular environment of a porous structure can affect both the energy of sorption of carbon dioxide and the nature of the primary interactions that drive sorption. Uniquely, the authors report the synthesis of single-crystal MFM-300 in the VIII state, with which they can subsequently perform single-crystal-to-single-crystal oxidation with oxygen to yield an isostructural VIV state. Owing to the presence of μ_2 -OH groups in the 1D VIII chains, upon oxidation to VIV no additional charge-balancing units remain in the pores, as the μ_2 -OH units become bridging oxides. Importantly, both states were characterized by single-crystal XRD, IR, and EPR, suggesting stoichiometric presence of either the VIII or VIV state in the individual structures, as well as the complete conversion from the hydroxyl to the oxyl bridge. These structures were subsequently dosed with carbon dioxide and extensively probed with neutron diffraction and inelastic scattering to study how these subtle changes in the chemical functionality of the pores affects sorption. Using these techniques, the authors find that in the VIII(μ_2 -OH) state carbon dioxide sorption and ordering in the pores is dominated by a strong hydrogen bonding interaction with the acidic hydroxyl proton. In contrast, in the VIV(μ_2 -O) state no such interaction exists, thus sorption and order are determined by weak interactions with the organic linkers, as well as intermolecular dipole interactions. Additionally, spectroscopic data were supported by DFT calculations.

B) This work is novel and highly interesting. Specifically, there does not exist significant precedent for single-crystal-to-single-crystal oxidation of the metal-centers that support the MOF architecture while obtaining an isostructural second state.

C) The data presented is of high quality and is presented in a clear and legible fashion.

D) Data is treated appropriately.

E) This work is of high quality and experimental detail has been sufficiently detailed for potential reproduction.

F) As the core tenant of this work is the complete oxidation from the VIII state to the VIV, these oxidation states must be determined unambiguously. Although the bond lengths in the single crystals reported support these oxidation states, and EPR shows a signal for VIV, it is not beyond reasonable doubt that there could exist incomplete oxidation in the assigned MFM-300(VIV).

Specifically, as the VIII state is EPR silent, it does not seem that one can look at the spectrum obtained for MFM-300(VIV) and conclusively state that there is no VIII present. The authors should therefore perform XPS or XANES on these structures to determine the full distribution of oxidation states within the two assigned structures. From such experiments the unambiguous assignment of a fully VIII and a fully VIV state should be possible. Did the authors attempt to chemically reduce MFM-300(VIV) back to MFM-300(VIII)? If so, a comment on this in the manuscript would be worthwhile and of interest to the readers. If not, this should be attempted.

G) References are appropriate.

H) This manuscript is well written and appropriately contextualized.

Reviewer #4 (Remarks to the Author):

This is a report of post-synthetic redox modification of a vanadium MOF in which V-(OH)-V linkages are converted to V-O-V linkages via oxidation of V(III) to V(IV) without destruction of the crystal lattice. This is an extraordinary achievement that allows for the comparison of two very similar guest channels that differ primarily in the capacity to H-bond to guest molecules. In this case, CO₂ is shown to occupy more ordered positions in the V(III) MOF.

This review will consider primarily the single-crystal X-ray diffraction data, but will be incomplete due to the absence of an experimental section for these data. In contrast, there is a highly detailed experimental section the neutron powder data. From what I can see, the work appears to meet current standards as judged from the CHECKCIF forms. (Of course, the A alerts are permissible given that the void spaces are the key features of these structures.)

Others will be required to review other sections of this paper, but my overall assessment encourages publication in Nature Communications.

Reviewers' comments:

Reviewer 1

This is a very nice paper coming from highly recognized research groups. In the present paper, authors try to identify the adsorption sites for CO₂ in two iso-structural MOFs after a modification in the vanadium oxidation state (III or IV).

Theoretical calculation predict pore channels in both samples around 0.67 x 0.67 nm (cross-section), large enough to accommodate N₂ and/or CO₂. However, N₂ adsorption measurements at 77K clearly anticipate that this small molecule (kinetic diameter 0.36 nm) is not able to access the porosity in both materials. Even N₂ adsorption measurements at 273 K are affected by these kinetic restrictions, preferentially in sample MFM-300 (VIII) (for instance, the N₂ adsorption isotherm at 273 K – black symbol in Figure S8- exhibit an upward deviation above 0.5 bar anticipating these kinetic limitations). The presence of these kinetic restrictions to access the inner porous structure can also affect a rather similar molecule such as CO₂ (0.33 nm kinetic diameter) as can be appreciated by comparing the low and high pressure CO₂ measurements. Whereas at atmospheric pressure sample MFM-300 (VIII) exhibits a larger adsorption capacity compared to sample MFM-300 (VIV), at high pressures (20 bar) both samples (III and IV) are rather similar, thus confirming the problems of accessibility at 1 bar.

In summary, gas adsorption measurements clearly show that the pore size in these MOFs is smaller than theoretical predictions, these adsorption measurements being highly affected by kinetic restrictions.

We have revised the manuscript accordingly. We fully agree with the referee that pore size plays a key role in the gas adsorption measurements of these two materials. Indeed, the PSD analysis from the CO₂ isotherm indicates a pore size distribution between 5.2 and 5.4 Å, lower than that observed from X-ray crystal structure. In addition, the large quadrupole moment of CO₂ molecule can also affect the gas-MOF interaction significantly. Therefore, it is a combination of functional group, surface area, pore size/geometry and gas-MOF interaction that dominates the gas adsorption property. Additional discussion has been added to the manuscript to clarify this point.

As described above, CO₂ adsorption measurements at 273 K and 1 bar show clear differences depending on the vanadium oxidation state. Authors attribute this behaviour to the role of the –OH group in the supramolecular binding of CO₂, as predicted by DFT calculations. However, DFT calculation predict a complete occupation of these sites (CO₂I) at very low surface coverage, i.e. the effect of these hydroxyl groups in the CO₂ adsorption performance, if any, would be observed at very low relative pressures (e.g., p/p₀< 10⁻⁵). Unfortunately, these values are far from the pressures measured in Figure 3a (highly above low coverages).

We have revised the manuscript accordingly. A log scale view of the CO₂ adsorption isotherms has been added as Figure S10. The referee is correct that there is an absence of strong/sharp adsorption in the low pressure region, but this is entirely consistent with the soft binding interaction based upon the primary weak hydrogen bonding observed in this case. The DFT calculation can only deal with fully occupied molecules and confirms that the inter-molecular CO₂...CO₂ interaction also plays a key role in gas uptake, which is further confirmed by the Qst analysis.

The main experimental evidence about the role of the hydroxyl groups could come from INS. The comparison of the INS for samples III and IV (Figure 5b) clearly show important differences below 75 meV mainly due to the O-H vibration, torsion, bending and tunnelling modes. However, upon CO₂ adsorption this region is mainly unaffected (see for instance the

difference spectra in Figure 5d), the main difference being at 110 meV due to the inhibited C-H vibrational motion.

The referee is correct. Upon CO₂ loading into MFM-300(V^{III}), the INS peak at 47 meV shows notable change, indicating the interaction with the –OH group (Figure 5d). The reason for the larger difference this and the peak at 110 meV (corresponding to C-H vibration) is because stoichiometrically there are 6 C-H groups for every one –OH group. INS signals are simply proportional to the number of hydrogen atoms present, and therefore, statistically C-H peaks will show a larger signal response. Overall, the changes on the INS spectra in both regions are entirely consistent with the nature of the supramolecular binding in these systems.

In summary, although DFT calculation predict a promoting role of –OH groups in sample MFM-300 (VIII) for CO₂ adsorption, the presence of kinetic restrictions in the gas adsorption measurements, the lack of information at very low relative-pressures and the absence of a clear evidence in the INS measurements makes the final conclusion very speculative.

Additional discussion has been added to clarify the role of pore size and activated diffusion. There is clear evidence in the INS of changes to both O-H and C-H groups on addition of CO₂, as confirmed also by the other reviewers.

Reviewer

2

Schroder, Yang and coworkers report on the formation of a redox active V(III) metal organic framework containing bridging hydroxide ions which is transformed in its V(IV) counterpart upon exposition to O₂ at 150°C. The main point addressed in this report is the impact of the formation of hydrogen bonding interactions with quadrupolar carbon dioxide guest molecules. The results are indicative of a significant interaction of the CO₂ guest molecules with the hydroxide ions as deduced from end-on configuration of the CO₂ molecule with the hydroxide ion, as determined from neutron diffraction, and the high values of isosteric heat of adsorption. As shown in the results the oxidation of the V(III) centers to V(IV) gives rise to the deprotonation of the hydroxide ion to an oxide ion with a concomitant modification in the CO₂ binding.

The results are of interest to the MOF community and deserves publication.

A few remarks:
1) Is the V(III) to V(IV) oxidation complete? Is there any possibility of V(III)/V(IV) mixed valence state? Could this be confirmed by electronic spectroscopy and/or magnetic measurements?

We thank the referee for this interesting question. The oxidation state of the vanadium centers was confirmed primarily from X-ray and neutron structural analysis. Firstly, the V-O bond distances, coupled with bond valence sum calculations (3.03 for the V^{III}-MOF and 3.96 for the V^{IV}-MOF), were compared for both MOFs (Table 1). Secondly, a detailed analysis of the neutron diffraction data for the bare materials confirms the absence (with 0 occupancy) of the bridging H center in the V^{IV}-MOF (Figure S2). Therefore, both structural analyses based upon X-ray and neutron diffraction indicate complete oxidation. In light of this reviewer's comments, XPS spectra have been collected for desolvated MFM-300(V^{III}) and MFM-300(V^{IV}) (Figure S28). The result shows that the binding energy of vanadium center in MFM-300(V^{IV}) is slightly higher than that in MFM-300(V^{III}) (514.5 vs 513.6 eV), consistent with an increase in the oxidation state of the V center. Furthermore, no shoulder peak (associated with V^{III}) is observed in the XPS spectrum of MFM-300(V^{IV}), again consistent with a complete oxidation. The preliminary magnetic measurements of these two samples show complicated magnetic behaviour which will be analysed and reported separately since these data are outside the scope of the current report. Additional

discussion has been added to the manuscript and the new Figure S28 has been incorporated into the SI.

2) Is the MOF oxidation reversible?

This is again a very interesting question. The oxidised material MFM-300(V^{IV}) can be reduced back to MFM-300(V^{III}) using Na₂SO₃ in aqueous solution, with the colour of MFM-300(V^{IV}) changing from dark purple back to pale green, corresponding to the re-formation of MFM-300(V^{III}). PXRD data however confirms that some partial decomposition of the framework upon reduction takes place under these conditions. Therefore, a quantitative reduction of MFM-300(V^{IV}) back to MFM-300(V^{III}) has not yet been achieved. Additional discussion has been added to the main text on these points.

3) Minor point: the authors use the term hydroxy throughout the text. Should not be hydroxide a more correct term?

Hydroxide is the noun and hydroxyl/hydroxy the adjective. We have changed hydroxy to hydroxyl (as in alkyl) throughout the manuscript where the adjective is required.

Reviewer

3:

Below are responses according to the review criteria A-H. Overall, I recommend publication if the points below can be addressed.

A) In this work, the authors employ isostructural MFM-300(VIII) and (VIV) to explore, using neutron diffraction and inelastic scattering, how the supramolecular environment of a porous structure can affect both the energy of sorption of carbon dioxide and the nature of the primary interactions that drive sorption. Uniquely, the authors report the synthesis of single-crystal MFM-300 in the VIII state, with which they can subsequently perform single-crystal-to-single-crystal oxidation with oxygen to yield an isostructural VIV state. Owing to the presence of μ -2-OH groups in the 1D VIII chains, upon oxidation to VIV no additional charge-balancing units remain in the pores, as the μ -2-OH units become bridging oxides. Importantly, both states were characterized by single-crystal XRD, IR, and EPR, suggesting stoichiometric presence of either the VIII or VIV state in the individual structures, as well as the complete conversion from the hydroxyl to the oxyl bridge. These structures were subsequently dosed with carbon dioxide and extensively probed with neutron diffraction and inelastic scattering to study how these subtle changes in the chemical functionality of the pores affects sorption. Using these techniques, the authors find that in the VIII(μ -2-OH) state carbon dioxide sorption and ordering in the pores is dominated by a strong hydrogen bonding interaction with the acidic hydroxyl proton. In contrast, in the VIV(μ -2-O) state no such interaction exists, thus sorption and order are determined by weak interactions with the organic linkers, as well as intermolecular dipole interactions. Additionally, spectroscopic data were supported by DFT calculations.

B) This work is novel and highly interesting. Specifically, there does not exist significant precedent for single-crystal-to-single-crystal oxidation of the metal-centers that support the MOF architecture while obtaining an isostructural second state.

C) The data presented is of high quality and is presented in a clear and legible fashion.

D) Data is treated appropriately.

E) This work is of high quality and experimental detail has been sufficiently detailed for potential reproduction.

F) As the core tenant of this work is the complete oxidation from the VIII state to the VIV, these oxidation states must be determined unambiguously. Although the bond lengths in the single crystals reported support these oxidation states, and EPR shows a signal for VIV, it is not beyond reasonable doubt that there could exist incomplete oxidation in the assigned MFM-300(VIV). Specifically, as the VIII state is EPR silent, it does not seem that one can look at the spectrum obtained for MFM-300(VIV) and conclusively state that there is no VIII present. The authors should therefore perform XPS or XANES on these structures to determine the full distribution of oxidation states within the two assigned structures. From such experiments the unambiguous assignment of a fully VIII and a fully VIV state should be possible. Did the authors attempt to chemically reduce MFM-300(VIV) back to MFM-300(VIII)? If so, a comment on this in the manuscript would be worthwhile and of interest to the readers. If not, this should be attempted.

We have responded to these points above.

The oxidation state of the vanadium centers was confirmed primarily from X-ray and neutron structural analysis. Firstly, the V-O bond distances, coupled with bond valence sum calculations (3.03 for the V^{III}-MOF and 3.96 for the V^{IV}-MOF), were compared for both MOFs (Table 1). Secondly, a detailed analysis of the neutron diffraction data for the bare materials confirms the absence (with 0 occupancy) of the bridging H center in the V^{IV}-MOF (Figure S2). Therefore, both structural analyses based upon X-ray and neutron diffraction indicate complete oxidation. In light of this reviewer's comments, XPS spectra have been collected for desolvated MFM-300(V^{III}) and MFM-300(V^{IV}) (Figure S28). The result shows that the binding energy of vanadium center in MFM-300(V^{IV}) is slightly higher than that in MFM-300(V^{III}) (514.5 vs 513.6 eV), consistent with an increase in the oxidation state of the V center. Furthermore, no shoulder peak (associated with V^{III}) is observed in the XPS spectrum of MFM-300(V^{IV}), again consistent with a complete oxidation. The preliminary magnetic measurements of these two samples show complicated magnetic behaviour which will be analysed and reported separately since these data are outside the scope of the current report. Additional discussion has been added to the manuscript and the new Figure S28 has been incorporated into the SI.

The oxidised material MFM-300(V^{IV}) can be reduced back to MFM-300(V^{III}) using Na₂SO₃ in aqueous solution, with the colour of MFM-300(V^{IV}) changing from dark purple back to pale green, corresponding to the re-formation of MFM-300(V^{III}). PXRD data however confirms that some partial decomposition of the framework upon reduction takes place under these conditions. Therefore, a quantitative reduction of MFM-300(V^{IV}) back to MFM-300(V^{III}) has not yet been achieved. Additional discussion has been added to the main text on these points.

G) References are appropriate.

H) This manuscript is well written and appropriately contextualized.

Reviewer 4:

This is a report of post-synthetic redox modification of a vanadium MOF in which V-(OH)-V linkages are converted to V-O-V linkages via oxidation of V(III) to V(IV) without destruction of the crystal lattice. This is an extraordinary achievement that allows for the comparison of

two very similar guest channels that differ primarily in the capacity to H-bond to guest molecules. In this case, CO₂ is shown to occupy more ordered positions in the V(III) MOF. This review will consider primarily the single-crystal X-ray diffraction data, but will be incomplete due to the absence of an experimental section for these data. In contrast, there is a highly detailed experimental section the neutron powder data. from what I can see, the work appears to meet current standards as judged from the CHECKCIF forms. (Of course, the A alerts are permissible given that the void spaces are the key features of these structures.) Others will be required to review other sections of this paper, but my overall assessment encourages publication in Nature Communications.

We apologise for missing some experimental details on the analysis of single-crystal X-ray diffraction data. This has now been reported in the supplementary information with new references.

REVIEWERS' COMMENTS:

Reviewer #1 (Remarks to the Author):

The authors have properly answered/addressed the reviewer concerns. The effect of the pore size in the adsorption measurements, diffusional limitations at low relative pressures and the different sensitivity of INS to the number of H atoms in C-H and O-H groups have been adequately explained/modified in the revised manuscript. After all these changes, the conclusions are more strongly supported by the experimental evidences.

The manuscript can now be accepted for publication in Nature Communications.

Reviewer #3 (Remarks to the Author):

The authors have sufficiently addressed both my concerns and those of reviewer #2. The XPS data demonstrates that the oxidation from V(III) to V(IV) is indeed complete or very close to it, and the authors detail their attempts at reversibly reducing back to the V(III) state, as requested. I can now recommend this manuscript for publication.